# Metabolomics and machine learning technique revealed that germination enhances the multi-nutritional properties of pigmented rice

Rhowell Jr. N. Tiozon[1,2], Nese Sreenivasulu[1], Saleh Alseekh [2], Kristel June D. Sartagoda [1], Björn Usadel [3] & Alisdair R. Fernie [2✉]

Enhancing the dietary properties of rice is crucial to contribute to alleviating hidden hunger and non-communicable diseases in rice-consuming countries. Germination is a bioprocessing approach to increase the bioavailability of nutrients in rice. However, there is a scarce information on how germination impacts the overall nutritional profile of pigmented rice sprouts (PRS). Herein, we demonstrated that germination resulted to increase levels of certain dietary compounds, such as free phenolics and micronutrients (Ca, Na, Fe, Zn, riboflavin, and biotin). Metabolomic analysis revealed the preferential accumulation of dipeptides, GABA, and flavonoids in the germination process. Genome-wide association studies of the PRS suggested the activation of specific genes such as *CHS1* and *UGT* genes responsible for increasing certain flavonoid compounds. Haplotype analyses showed a significant difference ($P < 0.05$) between alleles associated with these genes. Genetic markers associated with these flavonoids were incorporated into the random forest model, improving the accuracy of prediction of multi-nutritional properties from 89.7% to 97.7%. Deploying this knowledge to breed rice with multi-nutritional properties will be timely to address double burden nutritional challenges.

[1] Consumer-driven Grain Quality and Nutrition Center, Strategic Innovation Platform, International Rice Research Institute, Los Baños 4030, Philippines. [2] Max-Planck-Institute of Molecular Plant Physiology, Am Mühlenberg 1, 14476 Potsdam-Golm, Germany. [3] IBG-4 Bioinformatics Forschungszentrum Jülich, Jülich, Germany. ✉email: fernie@mpimp-golm.mpg.de

Rice (*Oryza sativa L.*) is a major staple cereal grain for most of the global population. Brown rice is an unpolished whole grain with an intact outer bran layer, embryo, and endosperm. Dietary fiber, amino acids, phytosterols, phenolics, and γ-aminobutyric acid (GABA), among others, are known to be present in brown rice. However, a large body of evidence has found that these compounds are more abundantly present in pigmented rice varieties. Over the past decade, scientific research on pigmented rice has substantiated its superiority over milled white rice. While white rice is more popularly consumed than pigmented rice, the loss of both the bran and the embryo during the refining process subsequently leads to nutrient loss[1], which may expose heavily dependent communities to dietary deficiencies and noncommunicable illnesses[2].

Germination has been established as an inexpensive bioprocessing strategy to induce the enrichment of nutrients and its bioavailability in cereal and seeds. The transformation from a dormant to a living seed involves complex, coordinated molecular processes dependent on the concurrent expression of a number of genes and influenced by a number of environmental conditions[3]. The process reactivates metabolic pathways, resulting in the breakdown of stored proteins and carbohydrates and the synthesis and accumulation of an array of metabolites with diverse structures and abundance[4]. Compared to brown rice, germinated brown rice has a greater concentration of crude proteins, carbohydrates, phenolic compounds, γ-oryzanol, dietary fiber, and essential amino acids such as leucine, lysine, phenylalanine, valine, alanine, glycine, and GABA[5]. Taken together, germinated rice demonstrates more nutritional benefits than its quiescent counterpart. Many studies have proven that pigmented rice has higher bioactive compounds than non-pigmented rice[6]. However, there is scarce information on the comprehensive nutritional composition of pigmented rice sprouts (PRS).

Advances in powerful, efficient, and high throughput detection methodologies have led to the intensive application of various metabolomic platforms to decipher the metabolite profile of various cereal crops, including multiple rice varieties at different development stages[2,7]. However, linking metabolite accumulation to the biochemical pathways involved in the germination of genetically varied pigmented rice remains obscure, as does deploying appropriate breeding approaches to leverage this information in specific selection programs. Furthermore, models that use the multi-pronged nutritional qualities of minerals, vitamins, and other secondary metabolites to categorize a collection of PRS varieties into unique ideotypes have not been developed to date. The present work aimed to (1) provide a more comprehensive understanding concerning the identification, quantification, and dynamics of nutrients and bioactive compounds in PRS varieties through high-resolution mass spectrometry-based metabolomics, (2) correlate pigmented rice performance with metabolite expression during germination and relate those metabolites to underlying biochemical pathways, (3) identify potential genetic variants influencing diverse health-promoting metabolic targets using metabolite genome-wide association study (mGWAS), and (4) link the generated big data to predict the nutritional classes of diverse germplasms of pigmented rice and identify rice varieties or accessions with superior dietary composition and health benefits.

## Results

### Germination enhances the free phenolics and certain micronutrients.

The dietary properties of pigmented rice sprouts were investigated to evaluate the effects of the germination process. Figure 1a demonstrates that germinated pigmented rice samples contained elevated levels of free phenolic compounds. In particular, there was a significant increase in these compounds in variable purple samples. Concurrently, there is a decrease in bound phenolics upon germination. As shown in Fig. 1aiii–iv, both free and bound proanthocyanidins have decreased throughout the germinated samples. The significant decrease in proanthocyanidins (tannins) during germination, may account for the substantial increase in bioaccessibility of the minerals. Figure 1b and Supplementary Data 1 demonstrate that germination resulted in elevated levels of minerals, including Ca, Na, Zn, and Fe, throughout the colored rice samples. Other minerals were not substantially affected by germination, as illustrated in Supplementary Fig. 1.

Figure 1c displays the vitamins identified and quantified by targeted metabolomics. Among water-soluble vitamins, the riboflavin content of germinated rice increased dramatically. The average riboflavin content (mg/100 g) in PRS is 0.83 (range: 0.19–2.77). After germination, the average amount of biotin (μg/100 g) in colored rice rose to 0.14 (range: 0–3.00). In fact, the biotin content in variable purple rice sprouts significantly increased, with a few lines of germinated purple and red rice also exhibiting high biotin content All rice samples exhibited a decrease in pantothenic acid content upon germination, suggesting that it may be catabolized to facilitate germination. However, the exact mechanism through which this occurs remains unknown. In contrast, there were no significant changes in the 10-methyltetrahydrofolate concentration following germination. Figure 1c(v) demonstrates that when colored rice is germinated, the α-tocopherol content of variable purple samples decreases significantly. Among germinated samples, pigmented rice had a higher average alpha-tocopherol content than non-pigmented rice, whereas variable purple sprouts have excellent levels of this compound.

The phenolics and micronutrients induced by the germination process were used as input variables to develop models for predicting the multi-nutritional quality of PRS. The clustering (Fig. 2) generated four distinct groups with diverse multi-nutritional profiles: Cluster 1 ($n = 96$) consists of lines that are deficient in GABA, vitamins, and minerals; Cluster 2 ($n = 122$) consists of lines that are abundant in folate, pantothenic acid, alpha-tocopherol, and sodium; Cluster 3 ($n = 55$) consists of lines that are abundant in GABA, riboflavin, biotin, calcium, and iron; and Cluster 4 ($n = 16$) which demonstrated strong antioxidant capacity and high zinc and folate contents (Supplementary Fig. 2). To minimize overfitting, the dimensionality reduction via correlation filter was applied, reducing the number of input variables to 19 (Supplementary Fig. 3). The models were tuned, and the optimized hyperparameters were summarized in Supplementary Data 2. Supplementary Fig. 4 plot demonstrates the mean decrease accuracy of the variables. Ca, total proanthocyanidin content (TPAC), and folate exhibited the highest mean decrease accuracy implying their importance in classifying the samples. In this study, a random forest (RF) was deployed to classify the multi-nutritional components of PRS, which comprise metabolites, GABA, vitamins, minerals, and antioxidants. The generated model has an accuracy of 89.7%, indicating that it is sufficiently accurate to be used for the selection of PRS variants to predict multi-nutritional properties. The confusion matrix generated by RF (Fig. 2) showed high true positive rates (TPR) for most clusters, with the exception of Cluster 4 due to the smaller number of members in this group. Hence, the following section explores ways to further improve the accuracy of the model by incorporating genetic markers.

### Differentially accumulated metabolites in germinated and non-germinated rice samples.

Through the UPLC-Q-Exactive Orbitrap-MS, over 600 annotated metabolites belonging to 20 families

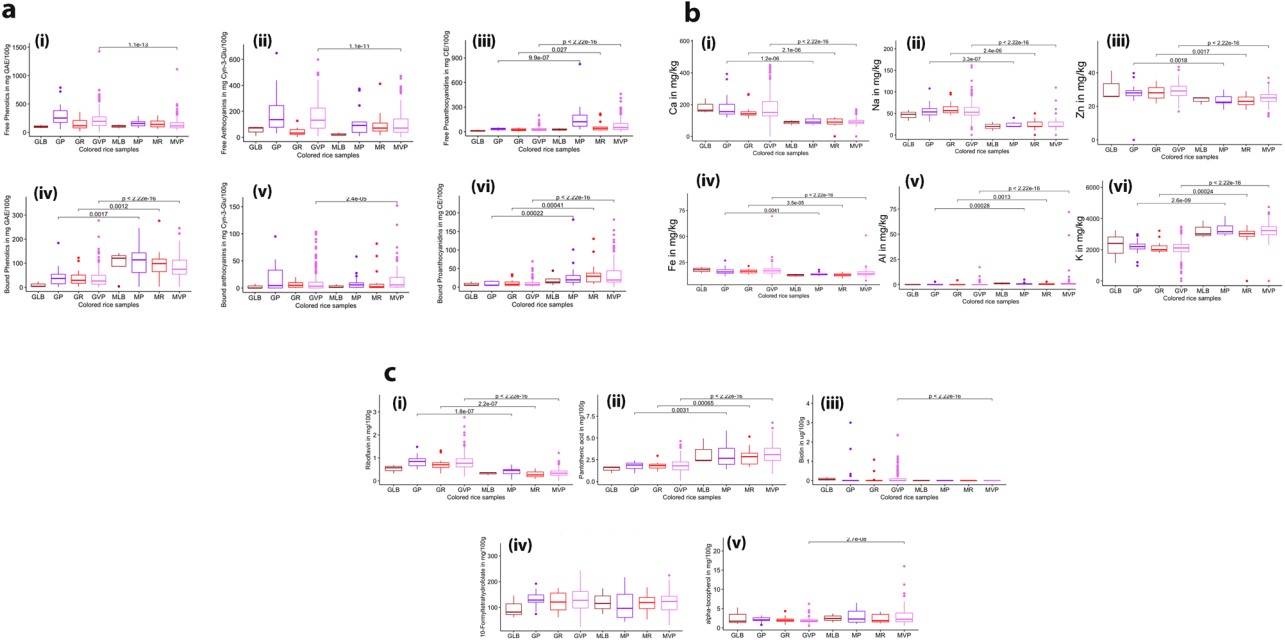

**Fig. 1 Germination alters different nutritional compounds. a** The process of germination enhanced free phenolics and free anthocyanins while lowering their bound counterparts (in mg/100 g rice): (i) free phenolics (ii) free anthocyanins, (iii) free proanthocyanidins, (iv) bound phenolics, (v) bound anthocyanins, and (vi) bound proanthocyanidins, **b** Germination process enhanced certain minerals (in mg/kg): (i) calcium, (ii) sodium, (iii) zinc, (iv) iron, (v) aluminum, (vi) potassium, **c** Some vitamins have increased after the germination process: (i) Riboflavin, (ii) Pantothenic acid, (iii) Biotin, (iv) 10-Formyltatrehydrofolate, (v) alpha-tocopherol. In the boxplot, the solid middle line depicts the median, while the lower and upper whiskers signify the 25th and 75th percentiles, respectively. (GLB Germinated Light brown, GP Germinated Purple, GR Germinated Red, GVP Germinated Variable Purple, MLB Matured (non-germinated) Light brown, MP non-germinated Purple, MR non-germinated Red, MVP non-germinated variable purple).

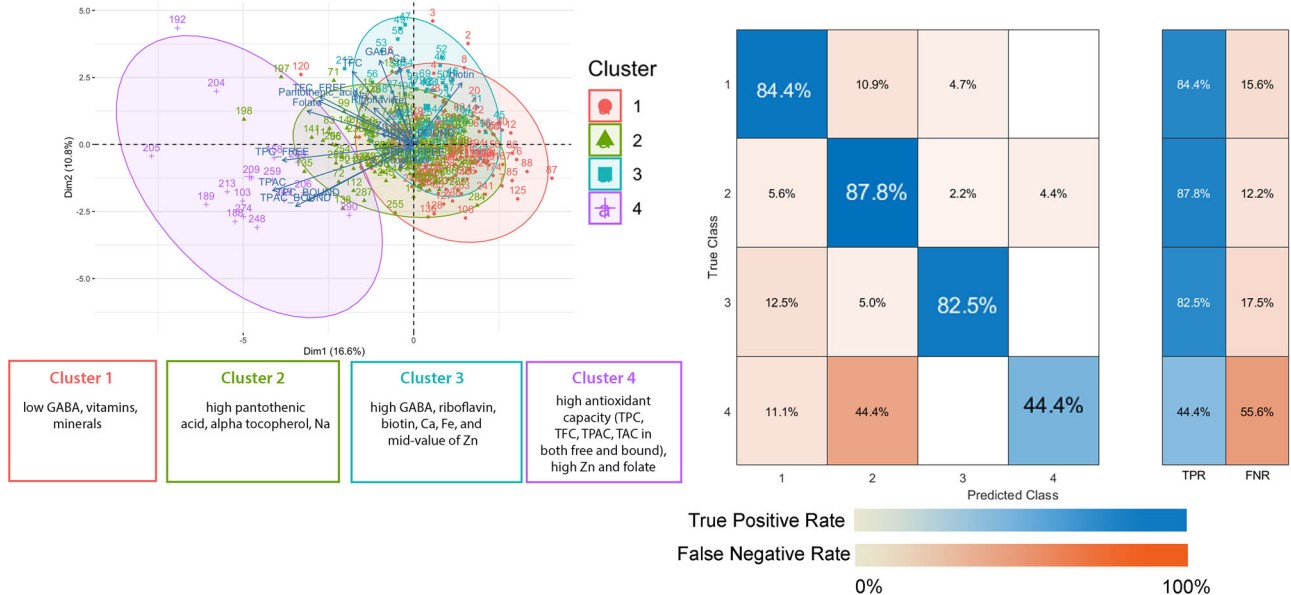

**Fig. 2 Random Forest model determined four clusters based on nutritional parameters.** TPR True Positive Rate, FNR False Negative Rate, GABA gamma amino butyric acid, Na sodium, Ca Calcium, Fe iron, Zn zinc, TPC Total Phenolic Content, TFC Total Flavonoid Content, TAC Total Anthocyanin Content, TPAC Total Proanthocyanidin Content. The confusion matrix consists of a blue gradient, which represents the true positive rate, and an orange gradient, which represents the false negative rates.

of chemicals were identified in the methanolic extract of PRS (Fig. 3a). The greatest portion of the metabolites was characterized in the groups of phenylpropanoids (38.99%), amino acids, peptides, and analogs (38.07%), and lipids and lipid-like molecules (10.78%). Figure 3b, c shows that the differentially regulated metabolites distinguished between germinated and non-germinated seeds. In examining the top differentially accumulated metabolites, amino acids and their derivatives obtained the highest VIP scores (Fig. 3d). It is noteworthy that many unknown peaks were clustered together with the group of peptides and flavonoids. Using the annotated metabolites in the pathway impact analysis unveiled significant alterations in butanoate,

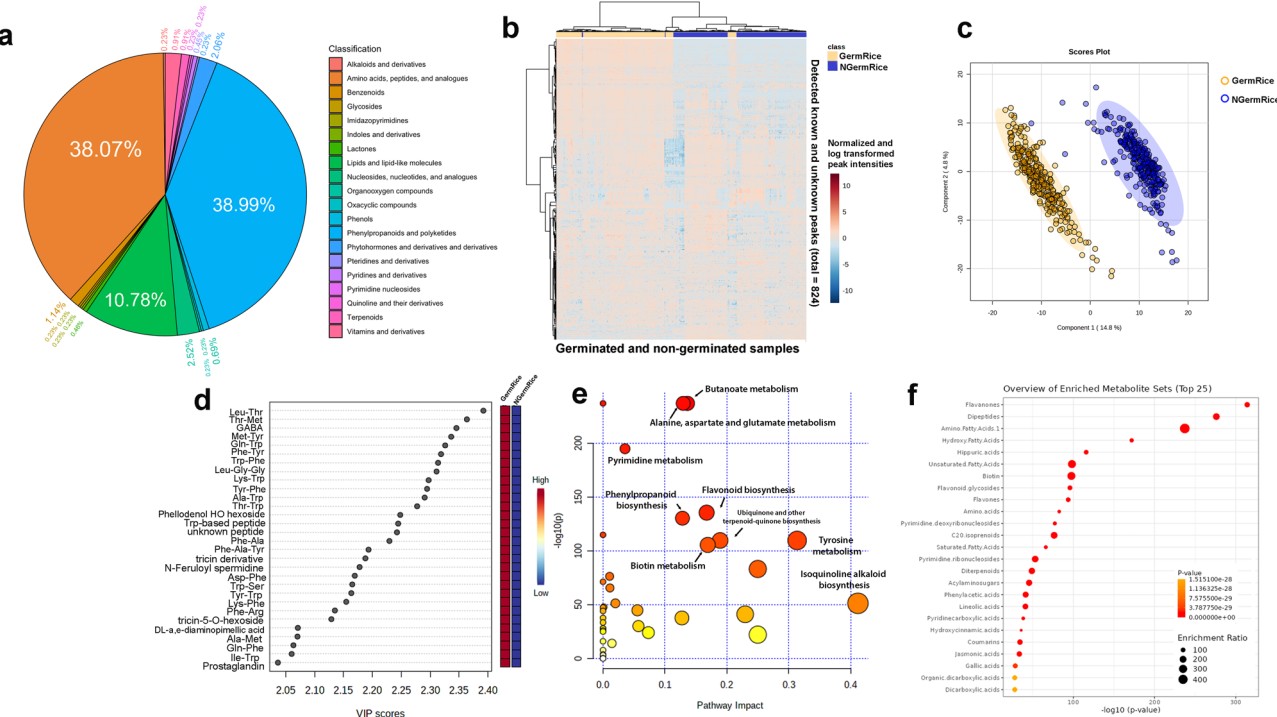

**Fig. 3 Metabolomic approach has shown differentially expressed metabolites between germinated (GermRice) and non-germinated (NGermRice) samples using the 80% MetOH extract. a** Classification of annotated metabolites found in germinated rice seeds, **b** heatmap based on 824 metabolites in which concentrations are normalized, log-transformed, and scaled (−10 to +10), **c** partial least squares-discriminant analysis (PLS-DA) plot that shows that 824 detected metabolites can distinguish between germinated and non-germinated groups. The gold-colored points are germinated samples while blue-colored points are non-germinated samples. The gold and blue shades show the group of samples classified in germinated and non-germinated samples, respectively. **d** Variable Importance in the Projection (VIP) scores from PLS-DA plot that shows the top important metabolites that distinguished germinated and non-germinated samples, **e** pathway impact analysis plot that identifies the main pathways significantly different upon the germination process, **f** enrichment analysis plot that demonstrated the subclass of compounds from the pathways that are significantly different between germinated and non-germinated samples.

flavonoid, and amino acid metabolism, including alanine, aspartate, tyrosine, and glutamate upon the germination process (Fig. 3e). Furthermore, the enrichment analysis of the main compound classes confirmed the presence of dipeptides and amino fatty acids, with notable increases in GABA and other aromatic amino acids observed upon germination (Fig. 3f). These findings highlight the dynamic metabolic changes that occur during the germination process and shed light on the enhanced nutritional composition of the germinated rice samples. Germinated variable purple rice had a broad range of GABA levels (0.09–80.24 mg/100 g rice), while red rice samples had the highest average content (14.46 mg/100 g rice) (Supplementary Fig. 5). Correlation analysis done on this work corroborates the accumulation of GABA in PRS as there is a strong positive correlation between GABA and Glu-Leu dipeptide levels (correlation = 0.95), as well as other dipeptides containing Glu units (Supplementary Data 3).

Figure 4 indicates that various flavonoid biosynthetic pathways may potentially be activated during the germination process. The germination process resulted in the production of higher naringenin chalcone serving as an intermediate of flavonoid compounds. Upstream of naringenin chalcone, flavones, such as apigenin and tricin derivatives, accumulate in the germinated seeds. Germination demonstrated a preferential increase in p-coumaric acid over cinnamic acid as intermediates in the formation of phenolic compounds (Supplementary Fig. 6). Consistent with this notion, downstream compounds from p-coumaric acids, such as caffeic acid and ferulic acid, are also upregulated. In fact, ferulic acid was found to be elevated on

average by 8-fold in germinated samples, particularly in purple and variable purple (Supplementary Fig. 5). This considerable increase may be attributed to the liberation of some ferulic acids from their bound form, which is consistent with the previously mentioned findings and other brown rice germination experiments[7]. Concurrently, ferulic-derived glycosides, such as feruloyl glucoside and feruloyl hexoside, have increased with germination. Furthermore, other flavonoid glycosides such as isorhamnetin-3-O-glucoside, quercetin-3-D-galactoside, and kaempferol rutinoside rose upon germination (Fig. 4 and Supplementary Fig. 5).

**The genetic markers from induced flavonoids and machine learning to classify dietary properties of rice sprouts**. Single-locus and multi-locus genome-wide association studies (GWAS) using a set of 558,526 high-quality biallelic single nucleotide polymorphism (SNP) markers resulted in identifying potential key genes influencing the production of specific secondary metabolites preferentially regulated in germinating sprouts (Supplementary Fig. 7). This metabolite-GWAS (mGWAS) approach has identified 47 candidate genes for eight flavonoids which accumulated preferentially in the PRS (Fig. 5a). Among these genes, both single-locus and multi-locus GWAS revealed *OsUGT* (represented by LOC_Os06g18670) on Chromosome 6 as the gene of interest in Kaempferol 3-glucoside-7-rhamnoside (K3G7R). The allelic variation in the samples associated with this genetic region revealed that samples that constitute the "A" allele contain significantly greater average content of K3G7R over the

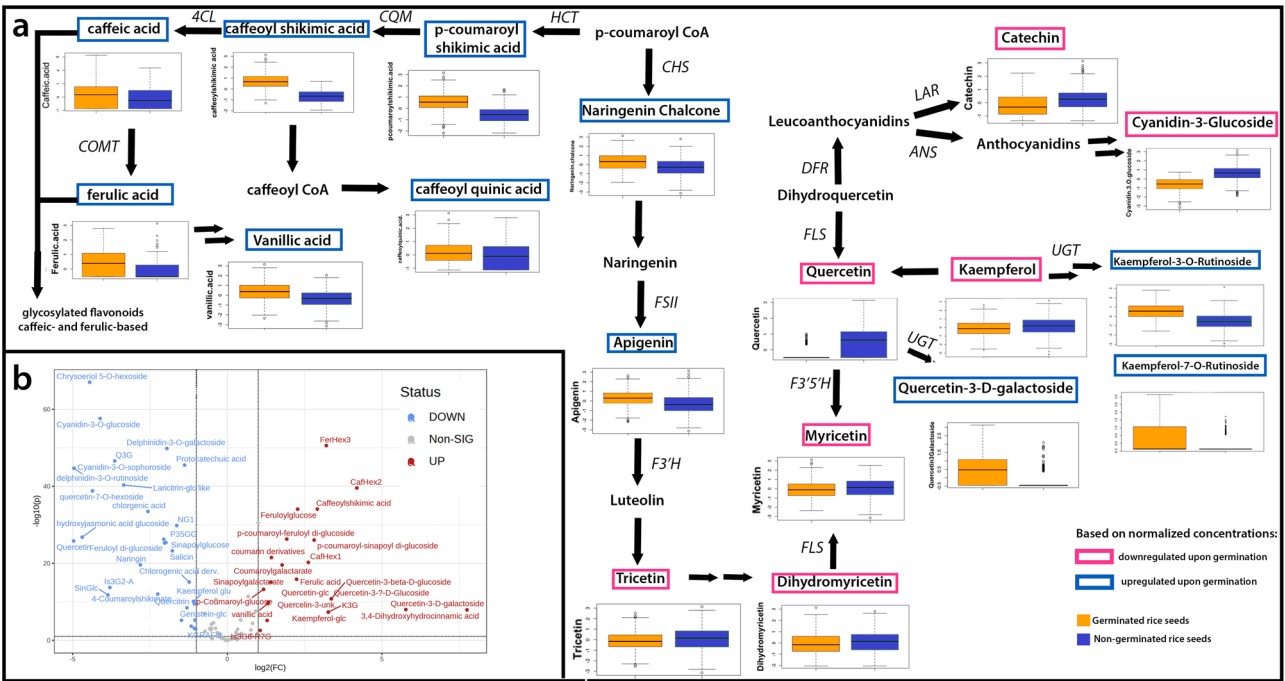

**Fig. 4 Phenolic metabolites induced and degraded during germination. a** Selected phenolic compounds in flavonoid biosynthesis pathway that were altered by the germination process. Concentrations of phenolic compounds were normalized, log-transformed, and scaled. In the boxplot, the solid middle line depicts the median, while the lower and upper whiskers signify the 25th and 75th percentiles, respectively. **b** Volcano plot of differentially expressed flavonoid and flavonoid glycosides. HCT shikimate o-hydroxycinnamoyltransferase, CHS Chalcone synthase, CQM 5-O-(4-coumaroyl)-D-quinate 3'-monooxygenase, 4CL 4-coumarate-CoA ligase, COMT caffeic acid O-methyltransferase, F3'5'H flavonoid 3',5'-hydroxylase, F3'H flavonoid 3'-hydroxylase, LAR leucoanthocyanidin reductase, ANS anthocyanin synthase, UGT Glucuronosyltransferase, FSII flavone synthase II, FLS flavonol synthase. Green-colored boxplots are the non-germinated seeds whereas red-colored boxplots are germinated seeds. The boxplot used the normalized concentrations of selected metabolites ranging -3 to +3 to better show distinction of germinated and non-germinated samples.

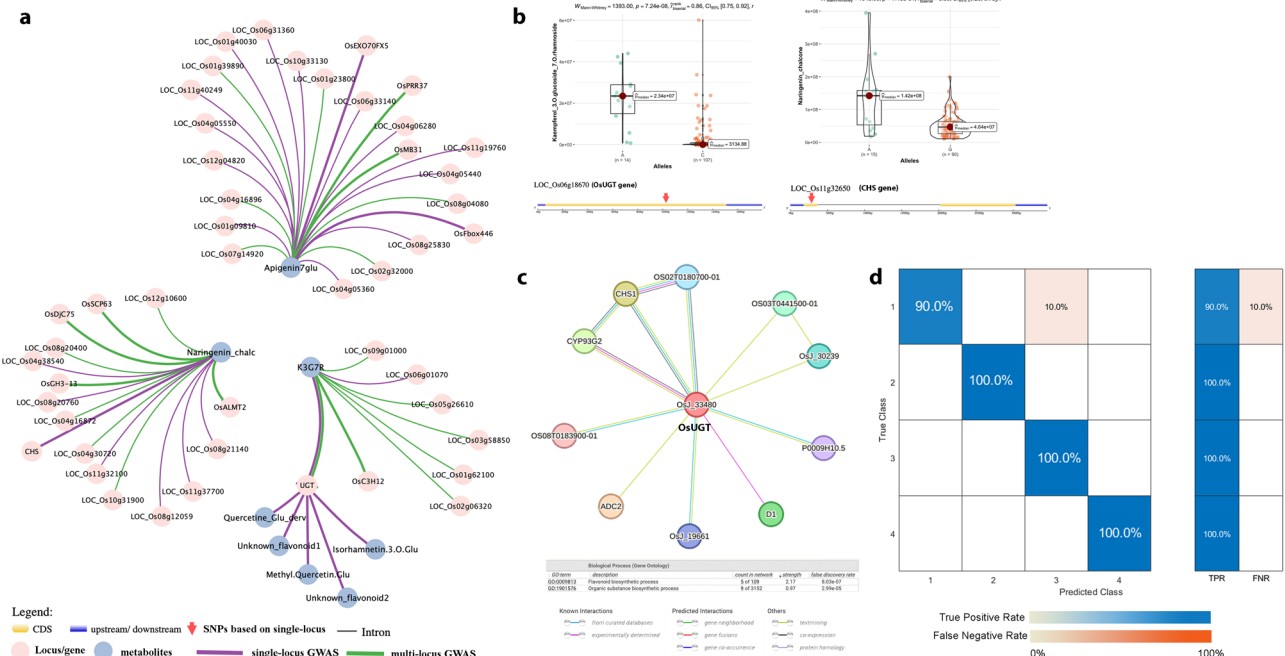

**Fig. 5 Genetic analysis of the metabolites induced by germination. a** Association network summary of candidate genes linked with the compounds induced by the germination process. The central nodes represent the metabolite of interest, and the smaller nodes are the associated loci and candidate genes, **b** boxplot showing the allelic variants from the genes of interest resulted from single-locus GWAS, **c** protein-protein interactions predicted for CHS1 and UGT genes, **d** improved Random Forest model incorporating the genetic markers. In the boxplot, the solid middle line depicts the median, while the lower and upper whiskers signify the 25th and 75th percentiles, respectively. *P*-values were obtained by Wilcoxon-Mann-Whitney-Test. The rank-biserial correlation (r) was used to present the effect size estimation. The confusion matrix consists of a blue gradient, which represents the true positive rate, and an orange gradient, which represents the false negative rates.

"C" allele (Fig. 5b). Further genetic analyses were conducted to examine the increase in naringenin chalcone levels after germination. Chalcone synthase 1 (CHS1) (represented by LOC_Os11g32650), resulted in the single-locus GWAS as the most significant gene (-log$_{10}$P = 10.21), which corresponds to its role in the synthesis and accumulation of naringenin chalcone. In fact, Fig. 5c demonstrates the interaction between CHS1 and OsUGT.

To test the hypothesis whether deploying these genetic markers identified from mGWAS will allow prediction of multi-nutritional properties of PRS, the genetic markers determined from the single-locus and multi-locus GWAS analyses of the metabolites were filtered before incorporating into the RF model. This model has the capability to classify a vast collection of germinated rice based on their dietary properties. Dimensionality reduction was accomplished by analyzing the linkage disequilibrium between the markers within the same genetic region and taking the correlation filter into account. The top genetic markers were then obtained by calculating their relative importance scores in the RF model (Supplementary Figure 8). The markers that were added to the model were OsUGT (LOC_Os06g18670), OsCH1 (LOC_Os11g32650), OsFbox446(LOC_Os08g3482), LOC_Os01g59780, LOC_Os09g17500, LOC_Os08g21140, OsCLE205 (S02_34577685), LOC_Os04g05360, and OsEX-O70FX5 (LOC_Os09g17810) in addition to the previous input variables such as GABA, vitamins, minerals, antioxidant component, and capacity. The overall accuracy of the RF model increased from 89.7% to 97.7% upon the incorporation of the genetic markers (Fig. 5d). More importantly, the superior cluster 4 group enriched for higher antioxidant property enriched with TPC, TFC, TPAC, and TAC true positive rate increased from 44.4% to 100%, conferring the importance of diagnostic markers of mGWAS targets.

## Discussion

Germination has been shown to induce new metabolites and enhance their bioavailability. Kim et al. (2020) demonstrated that the germination of brown rice induced compounds belonging to the class of acidic compounds, amino acids, sugars, and lipid metabolites[7]. GABA and dipeptides consisting mostly of essential amino acids such as tryptophan, phenylalanine, lysine, methionine, and threonine, were the most abundant metabolites in all germinated rice varieties. Similar outcomes were seen in germinating other cereals, such as wheat and triticale[8]. During germination, enzymatic hydrolysis of storage proteins initiates a reservoir of small peptides that are translocated to the growing embryo for nutritional supply. It has been shown that the transfer of these nutritive peptides by particular peptide transporters contributes to the growth and development of Arabidopsis and several monocots, such as rice[9]. Compared with non-pigmented samples, PRS had elevated levels of dipeptides constituting aromatic amino acids such as tyrosine, tryptophan, and phenylalanine. This observation may be attributed to the formation of soluble phenylpropanoids in the pigmented samples, as these aromatic amino acids play vital roles in the phenylpropanoid pathway[10]. The characterization and the possible role of these peptides may be elucidated by further proteomic analysis in PRS. For instance, quantitative shotgun proteomics of germinated brown rice revealed proteins associated with gibberellin signaling, protein trafficking, and the ABA-mediated stress response[11]. It is generally known that germination produces GABA, and its synthesis and role as a functional nutrient have been well established[5]. Various conditions, such as varying water content, pressure level, and incorporation of antimicrobial polymers, were tested to enhance the biosynthesis of GABA content in brown rice[5]. However, the GABA content of PRS has not been thoroughly explored. In the analyzed rice samples, PRS has a greater proportion of GABA than its non-pigmented equivalent. In tea plants, the metabolic flux to increase GABA is positively associated with genes involved in catechin biosynthesis[12]. Consequently, the buildup of GABA in red rice may be linked to its high catechin concentration. However, further explorations are necessary to validate the preferential accumulation of GABA in colored rice samples. In rice and higher plants, GABA is produced from L-glutamic acid (Glu) as it is catalyzed by glutamic acid decarboxylase through the GABA shunt pathway.

Certain phenolic compounds were also found to be differentially accumulated in the germinated seeds. For instance, the increase in flavones may be attributed to their presumed function as the initial line of defense against environmental stressors during germination, indicating that specific flavones may play a crucial role in the rice seed germination process as well as early seedling development. The preferential increase in p-coumaric acid over cinnamic acid can be attributed to the role of trans-cinnamate 4-monooxygenase in the conversion of these two compounds. The increase in p-coumaric acid brought about an increase in the coumarin-glycosylated compounds such as p-coumaroyl-feruloyl diglucoside and p-coumaroylglucoside, coumaroylgalactarate, and p-coumaroyl-sinapoyldiglucoside (Fig. 4b). The increase in the flavonoid glycosides can be attributed to the activation of UDP-glucuronosyltransferases (UGT) during the germination process. Although it is known that flavonoid glycosides accumulate in plants in response to abiotic stressors, their role in germination remains unclear. Despite this, flavonoid glycosides have been extensively researched for their beneficial pharmacological effects on human health[13]. On the other hand, the decrease in the aglycone moiety such as quercetin and kaempferol may be due to the overexpression of UGT favoring the production of their counterpart glycosylated flavonoids. Unsurprisingly, water-soluble anthocyanins such as cyanidin-3-O-glucoside, delphinidin-3-O-galactoside, and delphinidin-3-O-rutinoside decreased in the process (Fig. 4b), which can be attributed to their tendency to diffuse into the soaking water during germination. Anthocyanins are also light-sensitive and prone to degradation, which could possibly explain the decreased levels after the germination process. Besides the ability of flavanols such as catechin and epicatechin to dissolve in water, the downregulation of these compounds may also be beneficial for seed growth because, as demonstrated in other plant species, higher levels of these compounds may inhibit the process of seed germination[14]. Because catechin and epicatechin are the most prevalent monomers of tannins in rice, the decrease in these flavanols confirms the decrease in the bound form of condensed tannins (Fig. 1).

Numerous investigations on a variety of cereals and pulses have shown that germination increases the concentration of phenolic compounds[15]. For instance, the germination of brown rice enhanced its total phenolic and flavonoid content[16]. The observed increase in free phenolics may be ascribed to germination-activated enzymes capable of degrading storage macromolecules and liberating bound phenolics. Furthermore, the increase in the free phenolics may be linked to the synthesis of new phenolic compounds upon germination. These results corroborate the observed transformation of bound and free phenolics during germination reported in other cereals[15]. The impact of germination on proanthocyanidins has paradoxical effects on human health, as they are both antioxidants and anti-nutrients that inhibit the absorption of certain minerals. Jia et al. (2012) reported that proanthocyanidins inhibit seed germination by increasing abscisic acid levels in germinating plants[17]. Clearly, the reduction in proanthocyanidins is beneficial for the seed germination of colored rice.

Mineral concentration, solubility, and bioavailability may be affected by germination or pre-germination[18]. In finger millet and green gram, for instance, the germination process decreased zinc bioavailability while increasing iron concentration[19]. In contrast, earlier research showed that germination increased the zinc content of brown rice[20,21]. The significant rise in the divalent cations ($Ca^{2+}$, $Zn^{2+}$, and $Fe^{2+}$) is consistent with the fact that germination reduces the phytic acid concentration, which may chelate strongly with these minerals[21]. Tannins may also influence the bioavailability of minerals through the formation of complexes. Among the minerals, the K content of rice decreased upon germination, validating earlier observations[22,23]. In fact, reduced K levels lead to increase seed germinability and can be applied as a biomarker for the germination capacity of seeds[23].

Whole-grain rice contains an appreciable amount of vitamins with critical health functions[24]. The increase in some B-vitamers in various edible seeds was quantified and traced to germination. Shohag et al. (2012) reported that sprouting soybean and mungbean increased their folate content by 65–274% and 78–326%, respectively[25]. In addition, sprouting buckwheat increased its vitamin B1 and B6 levels to about 11.8 mg/100 g dry weight[26]. Similarly, the B1 and B6 vitamin content of brown rice surged following germination[27]. This finding indicates that a 100 g of riboflavin-enriched, pigmented rice sprouts is adequate to fulfill the current RDA for riboflavin in adults (1.0–1.7 mg/day)[24]. During seed development, riboflavin acts as a precursor of essential cofactors and plays a key role in coordinating cellular energy and cell cycle[28]. It has also been linked to higher levels of superoxide dismutase during seed germination, indicating its potential to scavenge superoxide anion radicals[29]. Biotin is a cofactor for various enzymes that catalyze carboxylation, decarboxylation, and transcarboxylation reactions in several vital metabolic processes. Wang et al. (2020) revealed that biotin could assist in the germination process of *Arabidopsis* under stress[30]. As shown in the succeeding sections, biotin is an essential cofactor for fatty acid synthesis, supporting the accumulation of important lipid compounds in PRS. However, the exact mechanism of biotin accumulation in PRS is yet to be deciphered. Nevertheless, the increase in biotin can only enhance the nutritional value of PRS. In fact, a 100-g portion of the biotin-enriched purple rice (var. WC 603) provides 10% of the recommended intake of biotin for adults (30 ug/day)[24]. Pantothenic acid, which is present in plants, has a role in the metabolic processes of glucose and fatty acid production through its function as a cofactor in the synthesis of ACP and CoA[31]. Previous studies have demonstrated that the impact of germination on folate content varies by plant type[25]. The ambiguous findings may be due to the fact that folate concentrations are highly dependent on the physiological state of the plants and their equilibrium within plant compartments[25]. Further examination of the influence of germination at various time points on all folate vitamers will shed light on the function of folate in the stages of germination in pigmented rice.

Tocopherols, widely known as vitamin E, are fat-soluble vitamins that comprise four isomers, namely, α-, β- δ-, and γ-tocopherols. The impact of germination on vitamin E content in seeds has been observed in other crops. For instance, germination has been shown to increase the levels of tocopherols and tocotrienols in grape seeds (*Vitis vinifera* L. cv. Albert Lavallée)[32], while decreasing the total vitamin E content of *Lupinus albus* L. var. Multolupa sprouts[33]. Although high tocopherol concentration is nutritionally beneficial, it must be regulated since it may diminish seed viability[34]. To understand the probable effect of germination and its possible mechanism, assessing the variations in levels of all vitamin E isoforms warrants further research. The selected nutritional parameters were input variables to build a classification model to predict their dietary classes. Random

Forest (RF) model can accurately predict clusters based on the multi-nutritional properties of PRS. Previously, we reported that RF could classify the antioxidant component and capacity of pigmented rice samples[6]. Herein, we utilized RF to cluster the samples based on the multi-nutritional properties. Cluster 1 represents inferior lines with respect to dietary properties as it has low GABA and micronutrient content. The rice lines in cluster 4 could serve as important donor lines as they have relatively higher antioxidant content and capacity, however, the TPR needs to be further increased to enhance the quality of the model. Targeted breeding programs may use the findings of cluster analyses as screening criteria to identify varieties with better multi-nutritional components. However, screening vast quantities of rice varieties manually is time-consuming and inefficient. Using an accurate classification model to predict a variety to a certain cluster has proven to be an effective approach for selecting and recommending superior lines for dietary purposes.

Genetic analysis of the differentially abundant metabolites in PRS sheds insights into the genetic regulation of these metabolites during rice germination. Flavonoid glycosides like kaempferol glycoside increased in PRS and the present study identified the importance of eight candidate genes (Table 1). The top gene for K3G7R which is *OsUGT* were also associated with other glycosylated flavonoids such as quercetin glucoside, methyl-quercetin glucoside, and isorhamnetin-3-glucoside, as well as two unknown flavonoids. Peng et al. (2017) revealed that *OsUGTs* influence the natural variation of rice flavones and may have a vital effect on rice stress tolerance[35]. Our findings confirmed that the upregulation of certain glycosylated flavonoids in PRS may potentially be attributed to the effect of *UGT* expression during the germination process. Besides the role of the UGT gene in catalyzing the glycosylation of most flavonoids, it also plays a major effect in grain pigmentation due to its contribution to anthocyanin and proanthocyanidin biosynthesis[36]. Besides *OsUGT*, single-locus GWAS revealed that *OsC3H12* (represented by LOC_Os01g68860) is involved in the germination and alteration of K3G7 content. The *OsC3H12* belongs to the family of *OsC3H* zinc finger proteins in the plant, which responds to ABA and GA during seed germination by altering the RNA metabolism of stress–responsive genes[37]. It can be surmised that K3G7 metabolite is one of the flavones released by the PRS in response to the alteration in the stress-responsive genes. However, further investigations are needed to confirm the mechanism. Chalcone synthase (*CHS*) catalyzes the primary committed step in rice flavonoid biosynthesis and is highly conserved across plant species[38]. In comparing the allelic variation in the *CHS1* gene, lines containing the "A" allele demonstrated a significantly higher average concentration of naringenin chalcone over the "G" allele (Fig. 5b). It can be surmised that the interaction of *OsUGT* and *CHS1* genes is vital for the induction of flavonoid glycosides during germination. Recently, Lam et al. (2022) demonstrated that rice mutants lacking the *CHS* genes exhibited a significant decrease in flavone and depleted tricin levels[38].

Multi-locus GWAS revealed more genes, such as *OsGH3-13* (LOC_Os11g32520) and *OsDjC75* (LOC_Os11g36520), which both play a role in plant growth and development and stress tolerance. However, a significant difference in naringenin chalcone levels among allelic variants were determined in *OsGH3-13* and not in *OsDjC75* gene (Supplementary Fig. 7c). Although these genes can be associated with seed germination, their relationship with naringenin chalcone warrants further analysis. Furthermore, downstream from naringenin chalcone, apigenin-7-glucoside's GWAS revealed genes associated with seed growth and development such as *OsFbox446* (LOC_Os08g34820) and *OsPRR37* (LOC_Os07g49460). The *OsFbox* protein-encoding genes modulate plant growth and various stages of seed

**Table 1 Information of selected genes identified in the genomic regions detected through the single-locus and multi-locus GWAS on the metabolites (Apigenin-7-glucoside, Kaempferol-3-glucoside-7-rhamnoside, and Naringenin Chalcone) induced by germination in pigmented rice sprouts.**

| Trait | Locus ID | Chr | LEAD SNP Position | ref | alt | p value | Gene symbol | Description |
|---|---|---|---|---|---|---|---|---|
| Apigenin-7-Glucoside | LOC_Os01g09810 | 1 | 5078855 | G | T | 1.76E−08 | - | - |
| | LOC_Os01g40030 | 1 | 22581132 | T | A | 1.76E−08 | - | - |
| | LOC_Os04g05360 | 4 | 2662922 | G | C | 1.76E−08 | - | - |
| | LOC_Os04g05440 | 4 | 2727335 | T | C | 1.76E−08 | - | - |
| | LOC_Os04g06280 | 4 | 3275377 | A | G | 1.76E−08 | - | - |
| | LOC_Os04g05550 | 4 | 2807081 | C | T | 1.76E−08 | - | MEGL10 - Maternally expressed gene MEG family protein precursor, expressed |
| | LOC_Os06g33140 | 6 | 19287351 | G | A | 1.76E−08 | - | - |
| | LOC_Os06g31360 | 6 | 18259969 | G | A | 3.24E−08 | - | - |
| | LOC_Os07g49460 | 7 | 29623625 | T | C | 6.22E−40 | OsPRRR37 | Heading date; influences yield and productivity |
| | LOC_Os08g25830 | 8 | 15718613 | T | G | 1.76E−08 | OsFbox446 | F-box protein 446 |
| | LOC_Os08g34820 | 8 | 21911476 | T | C | 1.76E−08 | | F-box-type E3 ubiquitin ligase X292 |
| | LOC_Os09g17810 | 9 | 10899658 | C | T | 1.76E−08 | OsEXO70FX5 | exocyst subunit EXO70 family protein FX5 |
| | LOC_Os10g33130 | 10 | 17356770 | G | A | 1.76E−08 | - | - |
| | LOC_Os11g40249 | 11 | 24011170 | A | A | 1.76E−08 | - | - |
| Kaempferol-3-glucoside-7-rhamnoside | LOC_Os01g60080 | 1 | 34755231 | A | G | 9.27E−08 | OsC3H12 | Zinc finger CCCH domain-containing protein 12; response to abscisic acid stimulus |
| | LOC_Os01g68860 | 1 | 40013976 | A | G | 1.86E−11 | OsUGT | C-pentosyl flavone pentosyltransferase anthocyanidin 3-O-glucosyltransferase; transferring hexosyl groups |
| | LOC_Os06g18670 | 6 | 10588880 | C | G | 1.38E−09 | | |
| Naringenin Chalcone | LOC_Os07g09830 | 7 | 5229282 | G | A | 6.14E−08 | - | - |
| | LOC_Os04g38540 | 4 | 22906086 | G | A | 2.91E−10 | - | - |
| | LOC_Os08g21140 | 8 | 12631828 | C | T | 6.21E−11 | - | - |
| | LOC_Os08g20760 | 8 | 12473075 | A | G | 1.26E−08 | | |
| | LOC_Os11g32520 | 11 | 19188841 | A | G | 3.81E−09 | OsGH3-13 | increased number of tillers enlarged leaf angles and dwarfism; response to auxin stimulus |
| | LOC_Os11g32650 | 11 | 19275716 | A | G | 4.60E−08 | OsCHS1 | Naringenin-chalcone synthase; seed development and flavone biosynthetic process |
| | LOC_Os11g32100 | 11 | 18977706 | A | C | 1.97E−09 | OsbHLH2 | basic helix-loop-helix protein 002; stress-tolerance |
| | LOC_Os11g37700 | 11 | 22288070 | T | G | 2.09E−09 | OsPDR3 | ABC transporter superfamily ABCG subgroup member 48 pleiotropic drug resistance 3 |

development in rice[39]. Analysis of the allelic variant in the *OsF-box446* gene revealed that allele "T" has significantly higher apigenin-7-glucoside levels compared with allele "C". In addition, multi-locus GWAS revealed OsPRR37 as one of the significant genes associated with apigenin-7-glucoside. This gene is known to influence the heading date, enhance the plant yield, and increase grain yield, with the relative expression of this gene is higher in the first 16 h of seed germination[40]. Correspondingly, the allelic variant in the OsPRR37 gene containing "T" has significantly higher apigenin-7-glucoside levels compared with allele "C" (Supplementary Fig. 7). Although flavones like apigenin-7-glucoside may have very specific functions in regulating plant development through their action in cell wall synthesis, the roles of these genes in the accumulation of Apigenin-7-Glucoside necessitates further investigation.

Machine learning enabled for more systematic and accurate genomic prediction of complex traits. The genetic markers that were incorporated along with other phenotypes enhanced the classification accuracy of the model. Furthermore, the prediction speed increased, and the training time decreased when the genetic markers were incorporated (Supplementary Data 2). The confusion matrix without the genetic markers (Fig. 2) revealed lower TPR for classifying the antioxidant component and capacity, whereas the incorporation of the genetic markers increased the TPR of Cluster 4, which consists of samples with high antioxidant component and capacity from 44.4% to 100% (Fig. 5d). Similarly, the TPR for Clusters 2 and 3 increased to 100%, while Cluster 1 increased from 84.4% to 90%. This approach may serve as a baseline model for future analyses incorporating additional features and algorithms that are anticipated to further improve the accuracy of the classification. Furthermore, it will likely be worthwhile to use these genetic markers for genomic predictions associated with the antioxidant content and capacity of rice. The result of this study provides important donor lines that can be an essential source of multi-nutrients and alleviate hidden hunger, especially in rice-consuming countries. Future studies combining the current approach with the network-based analysis[41] will likely provide further insights into the underutilized yet readily available source of nutrition.

## Methods

**Genetic material and growth conditions**. The diversity set ($n = 293$) of pigmented rice consisting of purple-colored ($n = 18$), variable-purple-colored ($n = 256$), and red-colored ($n = 16$) varieties and light brown ($n = 3$) were selected from the International Rice Research Institute (IRRI) and sown during the 2019 dry season at the experimental station of IRRI, Los Baños, Laguna, the Philippines under well-maintained, irrigated conditions (Supplementary Data 4). After harvest, the grains were collected, air-dried until they held 14% moisture, and dehulled to remove the inedible outer hull (Zaccaria PAZ-1/DTA testing rice mill, Brazil).

**Sample preparation and germination**. Dormancy was broken by incubating seeds at 50 °C for five days. Germination was carried out following Cáceres et al. (2017) with modifications[42]. Rice (50 grams) of each cultivar were washed with deionized water, surface sterilized with sodium hypochlorite 0.1% at 28 °C for 30 minutes, then rinsed with deionized water three times. Rice was dispersed on a Petri plate and soaked in deionized water (1:5, w/v) at 28 °C for 24 hours. Soaking water was discarded, and seeds were spread on Petri plates lined with moist laboratory paper and placed in the germination chamber (HiPoint Seed Germination Chamber, SG-650) for 48 hours set at 37 °C with a relative humidity of >90%. PRS was packed in

glassine bags and lyophilized for 72 hours at 1.1 bar (Christ LCG LYO Chamber Guard, Germany). Freeze-dried PRS was ground to a fine powder (using Mixer Mill MM400, Germany) and kept at −20 °C for further analysis.

**Genetic analysis**. The SNP marker data were generated by genotype-by-sequencing (GBS) and screened based on ≥90% call rate, locus homozygosity, and minor allele frequency (MAF) ≥ 0.05. Genome-wide association studies (GWAS) based on the mixed linear model for single-locus analysis and MLMM, BLINK, and FarmCPU models for multi-locus analysis were conducted using the R package of Genomic Association and Prediction Integrated Tool (GAPIT)[43]. Summary statistics was reported in Supplementary Data 5 and 6. Furthermore, the population structure estimation which includes the calculation of principal component analysis and kinship matrixes were also conducted using this software. The figures were then re-created using the using the rMVP (A Memory-efficient, Visualization-enhanced, and Parallel-accelerated tool) R package[44]. TASSEL 5.2.87 was used to convert the genetic markers to numerical genotype, and unknown entries were imputed using the Euclidean distance, estimated from the evaluation of the five nearest neighbors. Haplotype blocks were examined using the blocks function implemented in PLINK 1.9, and the Haploview program was used identify tag SNPs based on the threshold of the LD coefficient (*D'*) set to 0.8[45].

SNPs associated with a *P*-value of <0.05 were considered significant and were used to generate the haplotypes. Pairwise comparisons between alleles were based on the Mann-Whitney test and *t*-test using the "ggstatplot" package in R[46]. Cytoscape software was utilized to visualize the results of the GWAS. The predicted protein-protein interactions involving *CHS1* and *OsUGT* were determined using the STRING-DB v10.5 software. The interaction network was supported by experimental evidence, text-mining, co-expression data, and curated database[47].

**Mineral determination by using Inductively Coupled Plasma Optical Emission spectroscopy (ICP-OES) analysis**. For the ICP-OES analysis of elements in rice, the ground whole grain rice samples (0.600–0.625 g) were digested using 20 mL of 1% $HNO_3$. The resulting samples were then subjected to ICP-OES to determine the mineral content, following the method previously established for rice analysis[48]. Ten elements (Ca, Na, Zn, Fe, Al, K, P, Cu, Mg, and Mo) were quantified across a total of 586 rice samples, consisting of 293 germinated and 293 non-germinated rice samples.

**Spectrophotometric analysis for free, bound, and total phenolics**. The free and bound phenolic and flavonoid fractions were determined using our previously optimized method[49]. Briefly, the pH differential method was used to estimate the anthocyanin content, while the vanillin assay was used for proanthocyanin content. The absorbance was measured using a microplate reader (SPECTROstar Nano, BMG Labtech, Germany). Phenolic content was expressed as mg of gallic acid equivalents (GAE) per 100 g of sample, flavonoid, and proanthocyanin content as mg of catechin equivalents (CE) per 100 g of sample, and anthocyanin content as cyanidin-3-O-glucoside (Cyn-3-Glu) equivalents.

**Extraction procedure for metabolomic analyses**. Briefly, 50 mg of fresh material was extracted with 1.2 mL 80% methanol. Then, 2 μL were injected individually into the Acquity UPLC system using an RP C8 column and analyzed by MS[50]. The

samples were measured in positive and negative ionization modes. Selected metabolites were quantified using standards, and their concentrations were determined by following the calibration curve with an excellent coefficient of determination (at least $r^2 = 0.99$). This rigorous quantification process ensures accurate and reliable measurement of the targeted metabolites in the samples (Supplementary Data 7–10). The mass spectra were acquired using an Orbitrap high-resolution mass spectrometer: Fourier-transform mass spectrometer (FT-MS) coupled with a linear ion trap (LTQ) Orbitrap XL (ThermoFisher Scientific, https://www.thermofisher.com. Chromatograms and mass spectra were evaluated by using Chroma TOF 4.5 (Leco) and TagFinder 4.2 software. Metabolite data correlation was analyzed using the website MetaboAnalyst[51] and Expressionist Analyst 14.0.5 (Genedata, Basel, Switzerland) (https://www.genedata.com/products/expressionist).

**Metabolomic analysis, multivariate analyses, and mathematical modeling of metabolite changes during germination**. Multiple multivariate statistical methods were employed to investigate the variability of the metabolome pattern in germinated rice seeds. To this end, principal component analysis (PCA), partial least squares-discriminate analysis (PLS-DA), variable importance in projection scores, heatmap, and pathway enrichment analysis for data visualization were performed using MetaboAnalyst 4.0 software[51]. Classifications, clustering, and regression methods were performed using R (Version 3.3.2, released 2016) and Python (version 3.11, released 2022). AGNES Ward Clustering technique was employed to cluster samples using the nutritional phenotypes. Several machine learning techniques, including random forests and artificial neural networks, have been applied in the past to predict the classification of rice based on its antioxidant components and capacity[6]. In this study, random forest (RF) was used to classify the multi-nutritional properties of PRS, as it showed good accuracy in classifying antioxidant components[6]. The model was developed using MatLab (R2021b) and utilized metabolite and colorimetric data that had been filtered using a correlation filter with a tolerance of r = ±0.70. The data set ($n = 293$) was split into training (70%) and testing (30%) subsets.

**Statistics and reproducibility**. The spectrophotometric measurements of phenolics, anthocyanins, and proanthocyanidins were conducted in triplicates for validation. To ensure the reliability of mineral concentration measurements in the top lines, duplicate measurements were carried out. The Kruskal-Wallis test was employed to assess significant differences between germinated and non-germinated samples, as well as among samples based on their color. This statistical analysis facilitated the precise identification of any notable variations in mineral content among the rice samples. In the case of metabolomic analysis, selected samples were duplicated in every batch to ensure accuracy. Quality controls were implemented for all assays to guarantee the precision and robustness of the results. During model training, 10-fold cross-validation was employed, where the training set was randomly divided into training (90%) and validation (10%) sets. This process was repeated ten times using ten different validation subsets from the original training set[52]. The accuracy of the model was calculated, along with the true positive and negative rates of the classification process. Statistically significant differences among data sets were established using one-way analysis of variance (ANOVA) and Tukey's post hoc test (Supplementary Data 11 and 12).

**Reporting summary**. Further information on research design is available in the Nature Portfolio Reporting Summary linked to this article.

## Data availability

Source data underlying Fig. 1 are provided in Supplementary Data 1. Source data for Fig. 3d–f and Fig. 4a are provided in Supplementary Data 10. Source data for Fig. 5b are provided in Supplementary Data 6. Additional data supporting the findings of this study are available from the authors upon reasonable request.

## Code availability

The workflow for the machine learning technique was compiled in Rhowell's github: https://github.com/Rhowell09/Machine-Learning-for-Rice-Nutritional-Components[52].

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

## Acknowledgements

R.N.T. acknowledges the Academy for International Agricultural Research (ACINAR) for funding his Ph.D. ACINAR, commissioned by the German Federal Ministry for Economic Cooperation and Development (BMZ), is being carried out by ATSAF (Council for Tropical and Subtropical Agricultural Research) e.V. on behalf of the Deutsche Gesellschaft für Internationale Zusammenarbeit (GIZ) GmbH. N.S. thanks the UK Biotechnology and Biological Sciences Research Council UK Research & Innovation program (Project BB/T008873/1) and the CGIAR Research Initiative on Market Intelligence for their financial assistance. We appreciate the efforts of Edwige G. N. Mbanjo and Tobias Kretzschmar for identifying the pigmented rice core collection and seed multiplication and technical support of Roldan Ilagan Edwige G. N. Mbanjo and Saurabh Badoni, for growing the diversity population in the field and greenhouse. We thank Socorro L. Carandang for isolating DNA from these samples and duly acknowledge the genotyping by sequencing services (GBS) of The Elshire Group Limited. We thank Gopal Misra for processing the GBS data for creating genotyping file. We also thank Anna Natoza and Jazlyn Uy for processing the starch structure data. R.N.T. would like to thank Dr. Damien Platten, Socorro Carandang, Janine Kaye Vitto, Nobe Atanacio, and Dennis Villegas for their assistance in the germination and extraction processes. Furthermore, R.N.T. would like to acknowledge Reuben James Buenafe and Annabella Klemmer for their assistance in programming-related matters. Lastly, R.N.T. would like to thank Mikaela Zoe Tiozon for her overall support.

## Author contributions

R.N.T., N.S., and A.F. conceptualized the manuscript. R.N.T. and S.A. performed the metabolomic analysis, R.N.T. and K.S. performed the measurement of bioactives through spectrophotometric methods. R.N.T. and N.S. analyzed the micronutrient data. R.N.T. and B.U. performed and analyzed the GWAS results. R.N.T. deployed the models. N.S., S.A., and A.F. supervised the project.

## Funding

## Competing interests

The authors declare no competing interests.
