## [Peer Review File · Communications Biology]

Reviewers' comments:

Reviewer #1 (Remarks to the Author):

The manuscript by Tiozon and co-authors tries to investigate dietary compounds and micronutrients during germination of pigmented rice sprouts (PRS). The authors focused on the diversity set of PRS, and high-resolution mass spectrometry-based- and targeted-metabolomics measured over 600 annotated metabolites as well as some phenolics and micronutrients. Machine learning approaches, including RF, were used for developing models to predict the multi-nutritional quality of PRS. The generated model exhibited an accuracy of 89.7%. After that, the metabolite-GWAS also showed strong candidate genes responsible for the formation of flavonoid glycosides. Combined with their genetic markers, the model was improved. Although the metabolomic approaches and the data interpretation are of overall good quality, the authors did not provide some critical information to support their points. For example, they mentioned "various flavonoid biosynthetic enzymes are activated during the germination process" (Fig. 4). How did you measure their enzymatic activity? I also had a concern about the reproducibility.

Specific comments:

Methods

- 1) Would you share the details of the diversity set, please? What are the varieties' name?
- 2) There was no explanation about identifying differentially expressed (regulated) metabolites (Fig. 3).
- 3) I did not find any program code for mathematical modeling of metabolite changes during germination. Could you please share all the code? I looked at <https://github.com/Rhowell09/Machine-Learning-for-Rice-Nutritional-Components>. Could you share all input data in the code, please? For example, BCR_models_only.csv and MBCR_cluster_add.summarypercluster.csv.

Figures and Tables

- 1) Overall, the resolution of some figures is too low. For example, Fig. 1, Fig. 3F, boxplots in Fig. 4A, and Fig. 5B.
- 2) Fig. 1: Would you please add information about the statistical tests and experimental replicates?
- 3) Fig. 1B (v): What is y-axis?
- 4) Fig. 2: I cannot follow the construction of PCA-biplot. How do you classify the four clusters?
- 5) Supplementary Figure 2: What is x-axis?
- 6) Fig. 3B: I do not understand that detected metabolites are $n = 824$. Please clarify it.
- 7) Fig. 3E: The main text does not mention this figure.
- 8) Supplementary Figure 5: Would you please revise the truncated p-value representation on the top?
- 9) The authors should explain the details of the figures. For example, the pseudo-colors and the values in the map (Supplementary Figure 6).
- 10) Regarding boxplots in Fig. 4A, I would recommend that you focus on the significant changes in metabolite levels between germinated and non-germinated rice seeds. Did you perform any statistical tests?

Data availability

This study provides valuable information about nutritional components in pigmented rice varieties. It might be a good idea to share your raw metabolite data in community-approved repository. Regarding 600 annotated metabolites, would you please share the list with MSI-compliant annotation levels? The authors should report them according to Alseekh et al. Nat Methods (2021).

Reviewer #2 (Remarks to the Author):

The manuscript by Tiozon et al., "metabolomics and machine learning technique reveals that germination enhances the multi-nutritional properties of pigmented rice", is of interest, particularly in

understanding biological processes such as germination and its impacts on nutritional components of rice.

The manuscript is well written, the results are well presented.

The study reported in this manuscript is a metabolomics approach and use of ML to investigate the changes in the metabolism of pigmented rice in the germination process.

The authors report a preferential accumulation of flavonoid compounds in the germination process, supported by activation of genes in flavonoid (biosynthesis) pathways.

In addition to flavonoids, the authors report on other components, e.g., minerals and vitamins that were impacted by germination process.

Although the work is well researched, results well presented and discussed, providing great insights, there are few points that authors could improve on:

>>> The authors applied untargeted metabolomics approach, and this provided an opportunity to tap into a wide coverage of the measured metabolome. As depicted in Figure 3, the annotated metabolites were of different classes, not only flavonoids. The impacted biological pathways included other pathways. It would be nice to include this in the discussion, and mechanistic description of the germination process and its effects on nutritional components. This can also be included in the abstract - as the letter puts much more emphasis on flavonoids only.

>>> On Figure 1, in the figure legends, it would be nice (for the reader) to see the n=? and statistical test applied

>>> double-check typos (from the abstract throughout the paper)

Reviewer #3 (Remarks to the Author):

Tiozon et al have done an impressive study, painstakingly analyzing 600 rice cultivars for their nutritional profile after germination. They use extensive metabolic and elemental profiling to measure nutrients and use those to build classifiers of the lines for nutritional value.

They conduct genetic analysis to identify loci important for the production of the measured nutrients.

This dataset will be valuable for both researchers and breeders going forward.

I have three critiques:

1. There is no description of how the elemental nutrients were measured. If the elements were measured with ICP-MS, did the authors measure Cd and As. The presence of these toxic elements has a significant effect on the nutritional value of the food and should be considered in classifying desirable and undesirable cultivars for breeding.

2. The authors jump straight from identifying loci to assuming that they have identified genes. This is not true. While I agree with the authors that several of the genes that they have discovered are likely candidates for the causal alleles, they have not demonstrated this, despite the language used in the manuscript ("resulted in identifying key genes influencing the production of specific secondary metabolite" for example). For all of these loci, there will be multiple genes in the linkage window. These sections should be re-written to make this clearer. ^[1]_{SEPP}

3. The manuscript is extremely dense and detailed, making it hard to get through. I would advise the authors to try to trim back on some of the detail to highlight the key points they would like the reader to take away.

MAX-PLANCK-GESELLSCHAFT

Max-Planck-Institut
für Molekulare Pflanzenphysiologie

MPI für Molekulare Pflanzenphysiologie ♦ 14424 Potsdam ♦ Germany

Prof. Dr. Alisdair Fernie
Group Leader –Central Metabolism

Tel./Phon + 49 (0) 331-5 67 – 8211
Fax + 49 (0) 331-5 67 - 8408
E-mail: fernie @mpimp-golm.mpg.de

Dear Reviewers,

We are pleased to submit the revised manuscript titled "**Metabolomics and Machine Learning Technique Reveals that Germination Enhances the Multi-nutritional Properties of Pigmented Rice**" for your kind consideration for publication. In response to your valuable feedback, we have made significant revisions to the manuscript. To facilitate the review process, all the changes have been highlighted in red for easy tracking. Additionally, we have provided a detailed rebuttal response to address the reviewers' comments.

To the best of our knowledge, the result of this study provided new insights into the metabolomic and genetic variation among the pigmented rice sprouts.

Given that this subject may contribute to current breeding efforts, we are confident that it can likely pick the attention of your diverse audience in **Communications Biology – Nature Portfolio**. We look forward to your further comments on this original work.

Yours sincerely,

Alisdair Fernie

Metabolomics and machine learning technique reveals that germination enhances the multi-nutritional properties of pigmented rice

Rhowell Jr. N. Tiozon^{1,2}, Nese Sreenivasulu¹, Saleh Alseekh², Kristel June D. Sartagoda¹, Björn Usadel³, Alisdair R. Fernie²

¹Consumer-driven Grain Quality and Nutrition Center, Strategic Innovation Platform, International Rice Research Institute, Los Baños, 4030, Philippines

²Max-Planck-Institute of Molecular Plant Physiology, Am Mühlenberg 1, 14476, Potsdam-Golm, Germany

³IBG-4 Bioinformatics Forschungszentrum Jülich, Jülich, Germany

Referee expertise:

Referee #1: Plant metabolomics and genomics

Referee #2: Plant metabolomics, ML

Referee #3: Plant quantitative genetics, metabolomics

REVIEWERS' COMMENTS	AUTHORS' FEEDBACK
Reviewer #1	
The manuscript by Tiozon and co-authors tries to investigate dietary compounds and micronutrients during germination of pigmented rice sprouts (PRS). The authors focused on the diversity set of PRS, and high-resolution mass spectrometry-based- and targeted-metabolomics measured over 600 annotated metabolites as well as some phenolics and micronutrients. Machine learning approaches, including RF, were used for developing models to predict the multi-nutritional quality of PRS. The generated model exhibited an accuracy of 89.7%. After that, the metabolite-GWAS also showed strong candidate genes responsible for the formation of flavonoid glycosides. Combined with their genetic markers, the model was improved. Although the metabolomic approaches and the data interpretation are of overall good quality, the authors did not provide some critical information to support their points. For example, they mentioned “various	Thank you for your kind comments. We apologize for the confusion. The statement was changed to “Fig. 4 indicates that various flavonoid biosynthetic pathways may potentially be activated during the germination process.” (see lines 176 – 177) The reproducibility of the metabolomic data was ensured by employing standards, calibration curves, and quality controls for each run. Additionally, the machine learning codes

flavonoid biosynthetic enzymes are activated during the germination process” (Fig. 4). How did you measure their enzymatic activity? I also had a concern about the reproducibility.	underwent ten-fold cross-validation to guarantee consistent and similar results.
1) Would you share the details of the diversity set, please? What are the varieties’ name?	Details with the variety details were provided in Supplementary Table.
2) There was no explanation about identifying differentially expressed (regulated) metabolites (Fig. 3).	Separate section were constructed and improved about the differentially expressed metabolites.
3) I did not find any program code for mathematical modeling of metabolite changes during germination. Could you please share all the code? I looked at https://github.com/Rhowell09/Machine-Learning-for-Rice-Nutritional-Components. Could you share all input data in the code, please? For example, BCR_models_only.csv and MBCR_cluster_add.summarypercluster.csv.	The input file and output files were added to the GitHub page. Kindly refer to the link below. https://github.com/Rhowell09/Machine-Learning-for-Rice-Nutritional-Components The metabolite changes during the germination were visualized using MetaboAnalyst software.
Figures and Tables 1) Overall, the resolution of some figures is too low. For example, Fig. 1, Fig. 3F, boxplots in Fig. 4A, and Fig. 5B.	The figures were enhanced for better visualization. Furthermore, high-resolution TIFF files were provided.
2) Fig. 1: Would you please add information about the statistical tests and experimental replicates?	Thank you for your suggestion. The details on the experimental replicates and statistical tests were added in the method section.
3) Fig. 1B (v): What is y-axis?	The y-axis is the amount of minerals in germinated and non-germinated rice samples. The labels were enlarged for clarity.

4) Fig. 2: I cannot follow the construction of PCA-biplot. How do you classify the four clusters?	The four clusters were generated by using AGNES Ward Clustering technique. For better clarity, the process was stated in the method section. Furthermore, the codes for clustering were reported in the following link: https://github.com/Rhowell09/Machine-Learning-for-Rice-Nutritional-Components
5) Supplementary Figure 2: What is x-axis?	The x-axis represents the clusters used for the modelling. I modified the image for clarity.
6) Fig. 3B: I do not understand that detected metabolites are n = 824. Please clarify it.	Apologies for the confusion. For better clarity, we modified the label. By employing GeneData Expressionist Analyst 14.0.5 software, we conducted peak clustering and identified over 800 significant peaks between germinated and non-germinated rice samples. Subsequently, we classified 682 compounds by matching their identities and correlating the unknown peaks with known metabolites. Based on this refined dataset, we updated the heatmap to provide a more comprehensive visualization of the relevant peaks and their respective profiles.
7) Fig. 3E: The main text does not mention this figure.	Thank you for your keen observation. We have included this figure and the respective discussion in the main text. See lines 159 – 169.
8) Supplementary Figure 5: Would you please revise the truncated p-value representation on the top?	Thank you for your keen observation and suggestion. The figure was revised.
9) The authors should explain the details of the figures. For example, the pseudo-colors and the values in the map (Supplementary Figure 6).	A short explanation was placed in the description of the figure. Please see the revised figure and label in Supplementary Figure 6.

10) Regarding boxplots in Fig. 4A, I would recommend that you focus on the significant changes in metabolite levels between germinated and non-germinated rice seeds. Did you perform any statistical tests?	Thank you for your comment. The boxplots in Figure 4A were improved for better readability. Statistical test was conducted to deduce significant differences between germinated and non-germinated rice samples (see method section). In the discussion, we focused mainly on compounds with significant differences.
Data availability This study provides valuable information about nutritional components in pigmented rice varieties. It might be a good idea to share your raw metabolite data in community-approved repository. Regarding 600 annotated metabolites, would you please share the list with MSI-compliant annotation levels? The authors should report them according to Alseekh et al. Nat Methods (2021).	These details were included in the supplementary materials.
Reviewer #2 (Remarks to the Author):	
The manuscript by Tiozon et al., "metabolomics and machine learning technique reveals that germination enhances the multi-nutritional properties of pigmented rice", is of interest, particularly in understanding biological processes such as germination and its impacts on nutritional components of rice. The manuscript is well written, the results are well presented. The study reported in this manuscript is a metabolomics approach and use of ML to investigate the changes in the metabolism of pigmented rice in the germination process. The authors report a preferential accumulation of flavonoid compounds in the germination process, supported by activation of genes in flavonoid	Thank you for synthesizing our work.

(biosynthesis) pathways. In additional to flavonoids, the authors report on other components, e.g., minerals and vitamins that were impacted by germination process.	
Although the work is well researched, results well presented and discussed, providing great insights, there are few points that authors could improve on: >>> The authors applied untargeted metabolomics approach, and this provided an opportunity to tap into a wide coverage of the measured metabolome. As depicted in Figure 3, the annotated metabolites were of different classes, not only flavonoids. The impacted biological pathways included other pathways. It would be nice to include this in the discussion, and mechanistic description of the germination process and its effects on nutritional components. This can also be included in the abstract - as the letter puts much more emphasis on flavonoids only.	Thank you for your comment. Additional discussion was incorporated to also highlights other compounds such as dipeptides and GABA. See lines 159 – 169.
>>> On Figure 1, in the figure legends, it would be nice (for the reader) to see the n=? and statistical test applied	Thank you for your suggestion. The details on the experimental replicates and statistical tests were added in the method section.
>>> double-check typos (from the abstract throughout the paper)	Thank you for your keen observation. Typos were corrected.
Reviewer #3 (Remarks to the Author):	
Tlozon et al have done an impressive study, painstakingly analyzing 600 rice cultivars for their nutritional profile after germination. They use extensive metabolic	Thank you for your kind comments.

and elemental profiling to measure nutrients and use those to build classifiers of the lines for nutritional value. They conduct genetic analysis to identify loci important for the production of the measured nutrients. This dataset will be valuable for both researchers and breeders going forward.	
I have three critiques: 1. There is no description of how the elemental nutrients were measured. If the elements were measured with ICP-MS, did the authors measure Cd and As. The presence of these toxic elements has a significant effect on the nutritional value of the food and should be considered in classifying desirable and undesirable cultivars for breeding.	Thank you for your observation. We incorporated the method for mineral determination in the Method section. We used Inductively Coupled Plasma Optical Emission spectroscopy (ICP-OES) analysis to determine 10 elements such as Ca, Na, Zn, Fe, Al, K, P, Cu, Mg, and Mo. However, Cd and As are possible to be quantified using ICP-MS. Thank you for your suggestion, and we could explore it for future investigations. See lines 516 - 529
2. The authors jump straight from identifying loci to assuming that they have identified genes. This is not true. While I agree with the authors that several of the genes that they have discovered are likely candidates for the causal alleles, they have not demonstrated this, despite the language used in the manuscript (“resulted in identifying key genes influencing the production of specific secondary metabolite” for example). For all of these loci, there will be multiple genes in the linkage window. These sections should be re-written to make this clearer. [SEP]	Thank you for your comment. We modified the statement to, “Single-locus and multi-locus genome-wide association studies (GWAS) using a set of 558,526 high-quality biallelic single nucleotide polymorphism (SNP) markers resulted in identifying potential key genes influencing the production of specific secondary metabolites preferentially regulated in germinating sprouts” See lines 206 - 209.
3. The manuscript is extremely dense and detailed, making it hard to get through. I would advise the authors to try to trim back on some of the detail to highlight the key	Thank you for your comment. We tried our best incorporating all the comments of the reviewers while highlighting the key points.

points they would like the reader to take away.	
---	--

MAX-PLANCK-GESELLSCHAFT

Max-Planck-Institut
für Molekulare Pflanzenphysiologie

MPI für Molekulare Pflanzenphysiologie ♦ 14424 Potsdam ♦ Germany

Prof. Dr. Alisdair Fernie
Group Leader –Central Metabolism

Tel./Phon + 49 (0) 331-5 67 – 8211
Fax + 49 (0) 331-5 67 - 8408
E-mail: fernie @mpimp-golm.mpg.de

George Inglis, PhD
Senior Editor
Communications Biology

Dear Dr. George Inglis,

We are pleased to submit the revised manuscript titled "**Metabolomics and Machine Learning Technique Reveals that Germination Enhances the Multi-nutritional Properties of Pigmented Rice**" for your kind consideration for publication. In response to the valuable feedback from the reviewers, we have made significant revisions to the manuscript. To facilitate the review process, all the changes have been highlighted in red for easy tracking. Additionally, we have provided a detailed rebuttal response to address the reviewers' comments.

To the best of our knowledge, the result of this study provided new insights into the metabolomic and genetic variation among the pigmented rice sprouts.

Given that this subject may contribute to current breeding efforts, we are confident that it can likely pick the attention of your diverse audience in **Communications Biology – Nature Portfolio**. We look forward to your comments on this original work.

Yours sincerely,

Alisdair Fernie

Metabolomics and machine learning technique reveals that germination enhances the multi-nutritional properties of pigmented rice

Rhowell Jr. N. Tiozon^{1,2}, Nese Sreenivasulu¹, Saleh Alseekh², Kristel June D. Sartagoda¹, Björn Usadel³, Alisdair R. Fernie²

¹Consumer-driven Grain Quality and Nutrition Center, Strategic Innovation Platform, International Rice Research Institute, Los Baños, 4030, Philippines

²Max-Planck-Institute of Molecular Plant Physiology, Am Mühlenberg 1, 14476, Potsdam-Golm, Germany

³IBG-4 Bioinformatics Forschungszentrum Jülich, Jülich, Germany

Referee expertise:

Referee #1: Plant metabolomics and genomics

Referee #2: Plant metabolomics, ML

Referee #3: Plant quantitative genetics, metabolomics

REVIEWERS' COMMENTS	AUTHORS' FEEDBACK
Reviewer #1	
The manuscript by Tiozon and co-authors tries to investigate dietary compounds and micronutrients during germination of pigmented rice sprouts (PRS). The authors focused on the diversity set of PRS, and high-resolution mass spectrometry-based and targeted-metabolomics measured over 600 annotated metabolites as well as some phenolics and micronutrients. Machine learning approaches, including RF, were used for developing models to predict the multi-nutritional quality of PRS. The generated model exhibited an accuracy of 89.7%. After that, the metabolite-GWAS also showed strong candidate genes responsible for the formation of flavonoid glycosides. Combined with their genetic markers, the model was improved.	Thank you for your kind comments. We apologize for the confusion. The statement was changed to "Fig. 4 indicates that various flavonoid biosynthetic pathways may

Although the metabolomic approaches and the data interpretation are of overall good quality, the authors did not provide some critical information to support their points. For example, they mentioned “various flavonoid biosynthetic enzymes are activated during the germination process” (Fig. 4). How did you measure their enzymatic activity? I also had a concern about the reproducibility.	potentially be activated during the germination process.” (see lines 176 – 177) The reproducibility of the metabolomic data was ensured by employing standards, calibration curves, and quality controls for each run. Additionally, the machine learning codes underwent ten-fold cross-validation to guarantee consistent and similar results.
1) Would you share the details of the diversity set, please? What are the varieties’ name?	Details with the variety details were provided in Supplementary Table.
2) There was no explanation about identifying differentially expressed (regulated) metabolites (Fig. 3).	Separate section were constructed and improved about the differentially expressed metabolites.
3) I did not find any program code for mathematical modeling of metabolite changes during germination. Could you please share all the code? I looked at https://github.com/Rhowell09/Machine-Learning-for-Rice-Nutritional-Components. Could you share all input data in the code, please? For example, BCR_models_only.csv and MBCR_cluster_add.summarypercluster.csv.	The input file and output files were added to the GitHub page. Kindly refer to the link below. https://github.com/Rhowell09/Machine-Learning-for-Rice-Nutritional-Components The metabolite changes during the germination were visualized using MetaboAnalyst software.
Figures and Tables 1) Overall, the resolution of some figures is too low. For example, Fig. 1, Fig. 3F, boxplots in Fig. 4A, and Fig. 5B.	The figures were enhanced for better visualization. Furthermore, high-resolution TIFF files were provided.

2) Fig. 1: Would you please add information about the statistical tests and experimental replicates?	Thank you for your suggestion. The details on the experimental replicates and statistical tests were added in the method section.
3) Fig. 1B (v): What is y-axis?	The y-axis is the amount of minerals in germinated and non-germinated rice samples. The labels were enlarged for clarity.
4) Fig. 2: I cannot follow the construction of PCA-biplot. How do you classify the four clusters?	The four clusters were generated by using AGNES Ward Clustering technique. For better clarity, the process was stated in the method section. Furthermore, the codes for clustering were reported in the following link: https://github.com/Rhowell09/Machine-Learning-for-Rice-Nutritional-Components
5) Supplementary Figure 2: What is x-axis?	The x-axis represents the clusters used for the modelling. I modified the image for clarity.
6) Fig. 3B: I do not understand that detected metabolites are $n = 824$. Please clarify it.	Apologies for the confusion. For better clarity, we modified the label. By employing GeneData Expressionist Analyst 14.0.5 software, we conducted peak clustering and identified over 800 significant peaks between germinated and non-germinated rice samples. Subsequently, we classified 682 compounds by matching their identities and correlating the unknown peaks with known metabolites. Based on this refined dataset, we updated the heatmap to provide a more comprehensive visualization of the relevant peaks and their respective profiles.
7) Fig. 3E: The main text does not mention this figure.	Thank you for your keen observation. We have included this figure and the respective discussion in the main text. See lines 159 – 169.
8) Supplementary Figure 5: Would you please revise the truncated p-value representation on the top?	Thank you for your keen observation and suggestion. The figure was revised.

9) The authors should explain the details of the figures. For example, the pseudo-colors and the values in the map (Supplementary Figure 6).	A short explanation was placed in the description of the figure. Please see the revised figure and label in Supplementary Figure 6.
10) Regarding boxplots in Fig. 4A, I would recommend that you focus on the significant changes in metabolite levels between germinated and non-germinated rice seeds. Did you perform any statistical tests?	Thank you for your comment. The boxplots in Figure 4A were improved for better readability. Statistical test was conducted to deduce significant differences between germinated and non-germinated rice samples (see method section). In the discussion, we focused mainly on compounds with significant differences.
Data availability This study provides valuable information about nutritional components in pigmented rice varieties. It might be a good idea to share your raw metabolite data in community-approved repository. Regarding 600 annotated metabolites, would you please share the list with MSI-compliant annotation levels? The authors should report them according to Alseikh et al. Nat Methods (2021).	These details were included in the supplementary materials.
Reviewer #2 (Remarks to the Author):	
The manuscript by Tiozon et al., "metabolomics and machine learning technique reveals that germination enhances the multi-nutritional properties of pigmented rice", is of interest, particularly in understanding biological processes such as germination and its impacts on nutritional components of rice. The manuscript is well written, the results are well presented. The study reported in this manuscript is a metabolomics approach and use of ML to investigate the changes in the metabolism of pigmented rice in the germination process.	Thank you for synthesizing our work.

The authors report a preferential accumulation of flavonoid compounds in the germination process, supported by activation of genes in flavonoid (biosynthesis) pathways. In additional to flavonoids, the authors report on other components, e.g., minerals and vitamins that were impacted by germination process.	
Although the work is well researched, results well presented and discussed, providing great insights, there are few points that authors could improve on: >>> The authors applied untargeted metabolomics approach, and this provided an opportunity to tap into a wide coverage of the measured metabolome. As depicted in Figure 3, the annotated metabolites were of different classes, not only flavonoids. The impacted biological pathways included other pathways. It would be nice to include this in the discussion, and mechanistic description of the germination process and its effects on nutritional components. This can also be included in the abstract - as the letter puts much more emphasis on flavonoids only.	Thank you for your comment. Additional discussion was incorporated to also highlights other compounds such as dipeptides and GABA. See lines 159 – 169.
>>> On Figure 1, in the figure legends, it would be nice (for the reader) to see the n=? and statistical test applied	Thank you for your suggestion. The details on the experimental replicates and statistical tests were added in the method section.
>>> double-check typos (from the abstract throughout the paper)	Thank you for your keen observation. Typos were corrected.
Reviewer #3 (Remarks to the Author):	

Tlozon et al have done an impressive study, painstakingly analyzing 600 rice cultivars for their nutritional profile after germination. They use extensive metabolic and elemental profiling to measure nutrients and use those to build classifiers of the lines for nutritional value. They conduct genetic analysis to identify loci important for the production of the measured nutrients. This dataset will be valuable for both researchers and breeders going forward.	Thank you for your kind comments.
I have three critiques: 1. There is no description of how the elemental nutrients were measured. If the elements were measured with ICP-MS, did the authors measure Cd and As. The presence of these toxic elements has a significant effect on the nutritional value of the food and should be considered in classifying desirable and undesirable cultivars for breeding.	Thank you for your observation. We incorporated the method for mineral determination in the Method section. We used Inductively Coupled Plasma Optical Emission spectroscopy (ICP-OES) analysis to determine 10 elements such as Ca, Na, Zn, Fe, Al, K, P, Cu, Mg, and Mo. However, Cd and As are possible to be quantified using ICP-MS. Thank you for your suggestion, and we could explore it for future investigations. See lines 516 - 529
2. The authors jump straight from identifying loci to assuming that they have identified genes. This is not true. While I agree with the authors that several of the genes that they have discovered are likely candidates for the causal alleles, they have not demonstrated this, despite the language used in the manuscript (“resulted in identifying key genes influencing the production of specific secondary metabolite” for example). For all of these loci, there will be multiple genes in the linkage window. These sections should be re-written to make this clearer. [SEP]	Thank you for your comment. We modified the statement to, “Single-locus and multi-locus genome-wide association studies (GWAS) using a set of 558,526 high-quality biallelic single nucleotide polymorphism (SNP) markers resulted in identifying potential key genes influencing the production of specific secondary metabolites preferentially regulated in germinating sprouts” See lines 206 - 209.

3. The manuscript is extremely dense and detailed, making it hard to get through. I would advise the authors to try to trim back on some of the detail to highlight the key points they would like the reader to take away.	Thank you for your comment. We tried our best incorporating all the comments of the reviewers while highlighting the key points.

REVIEWERS' COMMENTS:

Reviewer #1 (Remarks to the Author):

The manuscript has much improved, and the authors did a good job to address the constructive suggestions of the reviewers.

Reviewer #3 (Remarks to the Author):

The authors have addressed several of my concerns and changed the specific sentence that I highlighted about identifying genes. However, they have not followed the logic of the comments for other sentences in the manuscript. The statements I think would still need to be changed are:

Abstract: "Genome-wide association studies of the PRS revealed the activation of specific genes such as CHS1 and UGT genes responsible for increasing certain flavonoid compounds."

I would change "revealed" to "suggested"

In results: "In fact, Fig. 5C demonstrates the interaction between CHS1 and OsUGT in flavonoid production."

I would remove the "in flavonoid production" Figure 5 demonstrates the interaction, not the involvement in flavonoid production.

MAX-PLANCK-GESELLSCHAFT

Max-Planck-Institut
für Molekulare Pflanzenphysiologie

MPI für Molekulare Pflanzenphysiologie ♦ 14424 Potsdam ♦ Germany

Prof. Dr. Alisdair Fernie
Group Leader – Central Metabolism

Tel./Phon + 49 (0) 331-5 67 – 8211
Fax + 49 (0) 331-5 67 - 8408
E-mail: fernie @mpimp-golm.mpg.de

Dear Reviewers,

We are pleased to submit the revised manuscript titled "**Metabolomics and Machine Learning Technique Reveals that Germination Enhances the Multi-nutritional Properties of Pigmented Rice**" for your kind consideration for publication. In response to your valuable feedback, we have made significant revisions to the manuscript. To facilitate the review process, all the changes have been highlighted in red for easy tracking. Additionally, we have provided a detailed rebuttal response to address the reviewers' comments.

To the best of our knowledge, the result of this study provided new insights into the metabolomic and genetic variation among the pigmented rice sprouts.

Given that this subject may contribute to current breeding efforts, we are confident that it can likely pick the attention of your diverse audience in **Communications Biology – Nature Portfolio**. We look forward to your further comments on this original work.

Yours sincerely,

Alisdair Fernie

Metabolomics and machine learning technique reveals that germination enhances the multi-nutritional properties of pigmented rice

Rhowell Jr. N. Tiozon^{1,2}, Nese Sreenivasulu¹, Saleh Alseekh², Kristel June D. Sartagoda¹, Björn Usadel³, Alisdair R. Fernie²

¹Consumer-driven Grain Quality and Nutrition Center, Strategic Innovation Platform, International Rice Research Institute, Los Baños, 4030, Philippines

²Max-Planck-Institute of Molecular Plant Physiology, Am Mühlenberg 1, 14476, Potsdam-Golm, Germany

³IBG-4 Bioinformatics Forschungszentrum Jülich, Jülich, Germany

Referee expertise:

Referee #1: Plant metabolomics and genomics

Referee #2: Plant metabolomics, ML

Referee #3: Plant quantitative genetics, metabolomics

REVIEWERS' COMMENTS	AUTHORS' FEEDBACK
Reviewer #1	
The manuscript by Tiozon and co-authors tries to investigate dietary compounds and micronutrients during germination of pigmented rice sprouts (PRS). The authors focused on the diversity set of PRS, and high-resolution mass spectrometry-based- and targeted-metabolomics measured over 600 annotated metabolites as well as some phenolics and micronutrients. Machine learning approaches, including RF, were used for developing models to predict the multi-nutritional quality of PRS. The generated model exhibited an accuracy of 89.7%. After that, the metabolite-GWAS also showed strong candidate genes responsible for the formation of flavonoid glycosides. Combined with their genetic markers, the model was improved. Although the metabolomic approaches and the data interpretation are of overall good quality, the authors did not provide some critical information to support their points. For example, they mentioned “various	Thank you for your kind comments. We apologize for the confusion. The statement was changed to “Fig. 4 indicates that various flavonoid biosynthetic pathways may potentially be activated during the germination process.” (see lines 176 – 177) The reproducibility of the metabolomic data was ensured by employing standards, calibration curves, and quality controls for each run. Additionally, the machine learning codes

flavonoid biosynthetic enzymes are activated during the germination process” (Fig. 4). How did you measure their enzymatic activity? I also had a concern about the reproducibility.	underwent ten-fold cross-validation to guarantee consistent and similar results.
1) Would you share the details of the diversity set, please? What are the varieties’ name?	Details with the variety details were provided in Supplementary Table.
2) There was no explanation about identifying differentially expressed (regulated) metabolites (Fig. 3).	Separate section were constructed and improved about the differentially expressed metabolites.
3) I did not find any program code for mathematical modeling of metabolite changes during germination. Could you please share all the code? I looked at https://github.com/Rhowell09/Machine-Learning-for-Rice-Nutritional-Components. Could you share all input data in the code, please? For example, BCR_models_only.csv and MBCR_cluster_add.summarypercluster.csv.	The input file and output files were added to the GitHub page. Kindly refer to the link below. https://github.com/Rhowell09/Machine-Learning-for-Rice-Nutritional-Components The metabolite changes during the germination were visualized using MetaboAnalyst software.
Figures and Tables 1) Overall, the resolution of some figures is too low. For example, Fig. 1, Fig. 3F, boxplots in Fig. 4A, and Fig. 5B.	The figures were enhanced for better visualization. Furthermore, high-resolution TIFF files were provided.
2) Fig. 1: Would you please add information about the statistical tests and experimental replicates?	Thank you for your suggestion. The details on the experimental replicates and statistical tests were added in the method section.
3) Fig. 1B (v): What is y-axis?	The y-axis is the amount of minerals in germinated and non-germinated rice samples. The labels were enlarged for clarity.

4) Fig. 2: I cannot follow the construction of PCA-biplot. How do you classify the four clusters?	The four clusters were generated by using AGNES Ward Clustering technique. For better clarity, the process was stated in the method section. Furthermore, the codes for clustering were reported in the following link: https://github.com/Rhowell09/Machine-Learning-for-Rice-Nutritional-Components
5) Supplementary Figure 2: What is x-axis?	The x-axis represents the clusters used for the modelling. I modified the image for clarity.
6) Fig. 3B: I do not understand that detected metabolites are n = 824. Please clarify it.	Apologies for the confusion. For better clarity, we modified the label. By employing GeneData Expressionist Analyst 14.0.5 software, we conducted peak clustering and identified over 800 significant peaks between germinated and non-germinated rice samples. Subsequently, we classified 682 compounds by matching their identities and correlating the unknown peaks with known metabolites. Based on this refined dataset, we updated the heatmap to provide a more comprehensive visualization of the relevant peaks and their respective profiles.
7) Fig. 3E: The main text does not mention this figure.	Thank you for your keen observation. We have included this figure and the respective discussion in the main text. See lines 159 – 169.
8) Supplementary Figure 5: Would you please revise the truncated p-value representation on the top?	Thank you for your keen observation and suggestion. The figure was revised.
9) The authors should explain the details of the figures. For example, the pseudo-colors and the values in the map (Supplementary Figure 6).	A short explanation was placed in the description of the figure. Please see the revised figure and label in Supplementary Figure 6.

10) Regarding boxplots in Fig. 4A, I would recommend that you focus on the significant changes in metabolite levels between germinated and non-germinated rice seeds. Did you perform any statistical tests?	Thank you for your comment. The boxplots in Figure 4A were improved for better readability. Statistical test was conducted to deduce significant differences between germinated and non-germinated rice samples (see method section). In the discussion, we focused mainly on compounds with significant differences.
Data availability This study provides valuable information about nutritional components in pigmented rice varieties. It might be a good idea to share your raw metabolite data in community-approved repository. Regarding 600 annotated metabolites, would you please share the list with MSI-compliant annotation levels? The authors should report them according to Alseekh et al. Nat Methods (2021).	These details were included in the supplementary materials.
Reviewer #2 (Remarks to the Author):	
The manuscript by Tiozon et al., "metabolomics and machine learning technique reveals that germination enhances the multi-nutritional properties of pigmented rice", is of interest, particularly in understanding biological processes such as germination and its impacts on nutritional components of rice. The manuscript is well written, the results are well presented. The study reported in this manuscript is a metabolomics approach and use of ML to investigate the changes in the metabolism of pigmented rice in the germination process. The authors report a preferential accumulation of flavonoid compounds in the germination process, supported by activation of genes in flavonoid	Thank you for synthesizing our work.

(biosynthesis) pathways. In additional to flavonoids, the authors report on other components, e.g., minerals and vitamins that were impacted by germination process.	
Although the work is well researched, results well presented and discussed, providing great insights, there are few points that authors could improve on: >>> The authors applied untargeted metabolomics approach, and this provided an opportunity to tap into a wide coverage of the measured metabolome. As depicted in Figure 3, the annotated metabolites were of different classes, not only flavonoids. The impacted biological pathways included other pathways. It would be nice to include this in the discussion, and mechanistic description of the germination process and its effects on nutritional components. This can also be included in the abstract - as the letter puts much more emphasis on flavonoids only.	Thank you for your comment. Additional discussion was incorporated to also highlights other compounds such as dipeptides and GABA. See lines 159 – 169.
>>> On Figure 1, in the figure legends, it would be nice (for the reader) to see the n=? and statistical test applied	Thank you for your suggestion. The details on the experimental replicates and statistical tests were added in the method section.
>>> double-check typos (from the abstract throughout the paper)	Thank you for your keen observation. Typos were corrected.
Reviewer #3 (Remarks to the Author):	
Tlozon et al have done an impressive study, painstakingly analyzing 600 rice cultivars for their nutritional profile after germination. They use extensive metabolic	Thank you for your kind comments.

and elemental profiling to measure nutrients and use those to build classifiers of the lines for nutritional value. They conduct genetic analysis to identify loci important for the production of the measured nutrients. This dataset will be valuable for both researchers and breeders going forward.	
I have three critiques: 1. There is no description of how the elemental nutrients were measured. If the elements were measured with ICP-MS, did the authors measure Cd and As. The presence of these toxic elements has a significant effect on the nutritional value of the food and should be considered in classifying desirable and undesirable cultivars for breeding.	Thank you for your observation. We incorporated the method for mineral determination in the Method section. We used Inductively Coupled Plasma Optical Emission spectroscopy (ICP-OES) analysis to determine 10 elements such as Ca, Na, Zn, Fe, Al, K, P, Cu, Mg, and Mo. However, Cd and As are possible to be quantified using ICP-MS. Thank you for your suggestion, and we could explore it for future investigations. See lines 516 - 529
2. The authors jump straight from identifying loci to assuming that they have identified genes. This is not true. While I agree with the authors that several of the genes that they have discovered are likely candidates for the causal alleles, they have not demonstrated this, despite the language used in the manuscript (“resulted in identifying key genes influencing the production of specific secondary metabolite” for example). For all of these loci, there will be multiple genes in the linkage window. These sections should be re-written to make this clearer. ^[1]_{SEP}	Thank you for your comment. We modified the statement to, “Single-locus and multi-locus genome-wide association studies (GWAS) using a set of 558,526 high-quality biallelic single nucleotide polymorphism (SNP) markers resulted in identifying potential key genes influencing the production of specific secondary metabolites preferentially regulated in germinating sprouts” See lines 206 - 209.
3. The manuscript is extremely dense and detailed, making it hard to get through. I would advise the authors to try to trim back on some of the detail to highlight the key	Thank you for your comment. We tried our best incorporating all the comments of the reviewers while highlighting the key points.

points they would like the reader to take away.	
---	--